# Mitogenomics of Chinch Bugs from China and Implications for Its Coevolutionary Relationship with Grasses

**DOI:** 10.3390/insects13070643

**Published:** 2022-07-17

**Authors:** Shujing Wang, Runqi Zhu, Huaijun Xue, Yanfei Li, Wenjun Bu

**Affiliations:** Institute of Entomology, College of Life Sciences, Nankai University, Tianjin 300071, China; wangsj@nankai.edu.cn (S.W.); zhurunqi1119@163.com (R.Z.); xuehj@nankai.edu.cn (H.X.)

**Keywords:** Blissidae, Poaceae, mitogenome, gene rearrangement, phylogeny

## Abstract

**Simple Summary:**

Blissidae is a group with high species richness in Lygaeoidea, and most of them live in the leaf sheaths of Poaceae plants. Here, 10 new mitogenomes from 10 species of eight genera from Blissidae were sequenced and analyzed. Gene rearrangement is only found in *Pirkimeru japonicus* (PiGXBS1), which is formed as the duplication of tRNA-H. Coupled with published data, phylogenetic analyses and divergence time were performed in Blissidae. The divergence within Blissidae began about 56 million years ago (Ma), in which the genus level divergence was concentrated at 30–51 Ma, slightly later than the diversification of Poaceae. The consistency of the divergence time between Blissidae and Poaceae might hint at the coevolutionary relationship between them. Our study provides a valuable resource for understanding insect–host relationships.

**Abstract:**

Blissidae (the Chinch bug) is a group with high species richness in Lygaeoidea, but there are only a few descriptions of mitochondrial genomes available. We obtained mitogenomes from 10 species of eight genera from Blissidae through second-generation sequencing technology. The length of the mitochondrial genome (excluding the control region) is between 14643 and 14385 bp; the content of AT is between 74.1% and 77.9%. The sequence of the evolution rate of protein coding genes was as follows: *ND5* > *ATP8* > *ND6* > *ND2* > *ND4* > *ND4L* > *ND1* > *ATP6* > *ND3* > *COIII* > *COII* > *CYTB* > *COI*. The mitogenomic structure of Blissidae is highly conservative. Gene rearrangement is only found in *Pirkimeru japonicus* (PiGXBS1), which is formed as the duplication of tRNA-H. The intergenic spacer between ND4 and tRNA-H, which form an obvious stem-and-loop structure, was found in all samples in this study. The phylogenetic trees generated by BI and ML indicated that Blissidae can be divided into three major clades: Clade A (only included *Macropes)*; Clade B ((*Pirkimerus* + *Bochrus*) + *Iphicrates*); and Clade C ((*Ischnodemus* + *Capodemus*) + (*Cavelerius* + *Dimorphopterus*)). The divergence within the Blissidae began at about 56 Ma. At the genus level, the divergence was concentrated at 30–51 Ma, slightly later than the diversification of Poaceae. The consistency of divergence time between Blissidae and Poaceae might hint at the coevolutionary relationship between them, but further molecular and biological evidence is still needed to prove it.

## 1. Introduction

Mitogenomes are widely used in the study of molecular evolution, phylogeny, population genetics, and phylogeography [1,2,3,4]. The insect mitogenome is a double-stranded, closed circular DNA molecule. It is generally between 14 and 20 kb in length, mainly influenced by the length of the control region (CR) [5,6]. Insect mitogenomes have conserved genetic composition and arrangement similar to that of *Drosophila yakuba* (Diptera: Drosophilidae) [7,8]. The information of the mitochondrial gene structure has been considered as the key signal of evolutionary biology [2,9,10,11,12].

Lygaeoidea is the second largest superfamily within Pentatomomorpha, currently comprising 14 families. In the published mitogenomes of Lygaeoidea, the gene structure is consistent with that of *Drosophila yakuba* [2,10,13,14]. As a group with relatively high species richness in Lygaeoidea, Blissidae has more than 420 species belonging to 55 genera, which are widely distributed in tropical and subtropical areas (Lygaeoidea Species File: http://lygaeoidea.speciesfile.org/Common/basic/Taxa.aspx?TaxonNameID=1208147, accessed on 1 June 2022). To date, nine genera of Blissidae: *Blissus* Burmeister, 1835; *Bochrus* Stål, 1861; *Capodemus* Slater and Sweet, 1972; *Cavelerius* Distant, 1903; *Dimorphopterus* Stål, 1872; *Iphicrates* Distant, 1903; *Ischnodemus* Fieber, 1837; *Macropes* Motschulsky, 1859; and *Pirkimerus* Distant, 1904, are recorded in China [15,16]. Almost all chinch bugs are sap-feeding and live in leaf sheaths, with their bodies showing significant morphological specialization and extreme flattening with the adaptation to the living environment [17,18,19,20,21]. Poaceae species are the main host plants of Blissidae. In previous study, through the comparison of the host plants utilized by chinch bugs and the plant phylogenies, it was hypothesized that the Blissidae species might be radiated and diversified in the Poaceae [22]. However, research on the molecular biology of chinch bugs is very scarce, and research on its internal phylogenetic relationship has never been carried out. The study of the internal phylogeny and evolutionary history of Blissidae will deepen our understanding of the family, and also provide important molecular data from species for the discussion of its relationship with host plants.

In this study, we obtained 10 species of eight genera from Blissidae (Appendix A) and analyzed the characteristics of the mitogenome of Blissidae. At the same time, the phylogeny and divergence time of Blissidae was preliminarily studied and its relationship with the host plants was discussed.

## 2. Materials and Methods

### 2.1. Sample Collection

We obtained 10 species of Blissidae in China (Appendix A). All specimens were preserved in 100% ethanol immediately in the field and stored at −20 °C before DNA extraction. The specimens were deposited in the Insect Molecular Systematics Lab, Institute of Entomology, College of Life Sciences, Nankai University, Tianjin, China (NKUM). The specimens were identified based on their morphological characteristics.

### 2.2. DNA Extraction, Sequencing, and Assembling

Genomic DNA was extracted from the mid-leg using a DNeasy Blood & Tissue Kit (QIAGEN). Sequencing was performed by Novogene (Tianjin, China) with an insert size of 250 bp and a pair-end 150 bp sequencing strategy on the Illumina platform. The mitogenomes were assembled in MitoZ [23] and IDBA-master [24] methods for assembly and mutual verification.

### 2.3. Annotation of Mitogenome

We use the Mitos Web Server (http://mitos.bioinf.uni-leipzig.de/index.py/, accessed on 1 June 2022) [25] to identify the boundaries of tRNA genes and the secondary structures of tRNAs. The start and stop codons of the protein-coding genes (PCGs) are determined by ORF Finder using invertebrate mitochondrial genetic codes, which are implemented by the NCBI website (https://www.ncbi.nlm.nih.gov/orffinder/, accessed on 1 June 2022). The rRNA boundaries are predicted by comparison with homologous regions of the other published lygaeoidea mitogenomes in GenBank. 

### 2.4. Phylogenetic Inference and Divergence Time Estimation 

We carried out phylogenetic analyses based on the mitochondrial dataset of 18 samples, including 10 samples of Blissidae obtained in this study, and eight outgroups were selected (four of them belonging to Lygaeoidea, and the other four belonging to Pyrrhocoridea) (Appendix A). Alignments of all PCGs with other mitogenomes were performed based on their amino acid sequences using MUSCLE implemented in MEGA X [26]. The rRNAs were aligned using MAFFT v7 with the option –G-INS-I [27]. All individual genes were then concatenated as a combined matrix. The PCG12RT matrix, including all 13 PCGs with third codon positions removed, two rRNA and all tRNA genes, was used for the final phylogenetic analysis.

The phylogenetic analyses were conducted utilizing Bayesian inference (BI) and maximum likelihood (ML). PartitionFinder 2 [28] was used to determine the optimal partitioning strategies and the best-fit nucleotide substitution model. The results of PartitionFinder for MrBayes v3.2.5 [29] and IQ-TREE v2 [30] revealed the best-fit nucleotide substitution model in Appendix A. Under the BI method, we used Phylobayes-MPI v.1.5a [31] and MrBayes 3.2.5 [29]. Regarding MrBayes analysis, a Markov chain Monte Carlo analysis (four chains) was run for 10,000,000 generations, and samples were recorded every 1000 generations. The first 25% of the samples were discarded as burn-in, and the remaining samples were used to summarize the Bayesian posterior probabilities (BPP). In the Phylobayes analysis, two independent Markov chain Monte Carlo chains were run after the removal of constant sites from the alignment and were stopped after the two runs had satisfactorily converged (maxdiff < 0.1). A consensus tree was computed from the remaining trees combined from two runs after the initial 25% trees of each run were discarded as burn-in. ML analyses of molecular dataset were conducted with IQ-TREE v2 [30], and the node support values were assessed by bootstrap resampling (BP) calculated using 1000 replicates. Finally, FigTree v 1.3.1 [32] was used to visualize the phylogenetic tree.

MCMCTree from the Phylogenetic Analysis by Maximum Likelihood (PAML) 4.9 package [33] was used to estimate divergence times in Blissidae with a relaxed molecular clock based on the ML tree of PCG12RT. The earliest fossil of Blissidae, *Eoblissus gallicus* (58.3–53.7 Ma), and the fossil of Berytidae, *Metacanthus serratus* (33.9–28.4 Ma), were used in the calculation. The main parameters during operation are as follows: model = 4, rgene_gamma = 2, 20. After a 20,000 burn-in, the MCMCTree program was run for 1,000,000 MCMC steps and sampled every 100. The robustness of the MCMCTree results was checked by comparing the consistency of at least two independent runs, with all parameters at least 200 for the effective sample sizes.

## 3. Results and Discussion

### 3.1. Mitogenome Comparison and Rearrangement

In the 10 mitogenomes obtained in this study from Blissidae, the length of the mitochondrial genome (excluding the control region) is between 14643 and 14385 bp; the content of AT is between 74.1 and 77.9% (Figure 1 and Figure 2). The nucleotide skew statistics for the whole genome obviously showed AT-skew and CG-skew; the AT content of rRNA was significantly higher than that of PCGs and tRNA (Table 1). For PCGs, the ratios of the nonsynonymous nucleotide changes (Ka) versus the synonymous nucleotide changes (Ks) were all below 1, indicating that they evolved according to the purifying selection [34]. The sequence of the evolution rate of the protein-coding genes was as follows: *ND5* > *ATP8* > *ND6* > *ND2* > *ND4* > *ND4L* > *ND1* > *ATP6* > *ND3* > *COIII* > *COII* > *CYTB* > *COI* (Appendix A).

The subregion of the intergenic spacer between the tRNA-H and the ND4 was found in all samples in our study (Figure 3A). Through structural prediction, we found that the subregion of the intergenic spacer formed a stem-and-loop structure. The mitogenome structure was consistent and identical to the putative ancestral arrangement of insects [35] among all of the species we obtained (expect the sample *Pirkimeru japonicus*), showing the highly conservative mitogenome structure of Blissidae. It is noteworthy that gene rearrangement of tRNA-H replication was found in *Pirkimeru japonicus* (PiGXBS1) (Figure 1, Figure 2, and Figure 3B). The second tRNA-H was located between tRNA-M and ATP8, and the distance between the two tRNA-H was large (Figure 2 and Figure 3B). Due to the consistent neck ring structure at the front ends of the two tRNA-H, we inferred that the rearrangement was due to the recent recombination caused by the stem-and-loop structure [10,36,37].

### 3.2. Phylogenetic Analyses and Divergence Time Estimation

The phylogenetic trees generated by BI and ML were highly consistent. The 10 species could be divided into three major clades: Clade A (only including *Macropes*); Clad B ((*Pirkimerus* + *Bochrus*) + *Iphicrates*); and Clade C ((*Ischnodemus* + *Capodemus*) + (*Cavelerius* + *Dimorphopterus*)) (Figure 4). The monophyly of *Macropes* was supported by the phylogenetic results.

The divergence within Blissidae began at about 56 Ma, and the divergence at the genus level was concentrated at 30–51 Ma. Previous research showed that Poaceae plants were the main hosts of the species included in the study, including Bambusoideae, Pooideae, Oryzoideae, Panicoideae, Chloridoideae, and Arundinoideae [22]. Recent studies revealed that the diversification within these subfamilies of Poaceae was about 66–54 Ma [38], slightly earlier than the diversification of species in Blissidae. The species of Blissidae live in leaf sheaths and are highly dependent on the host plants; we speculated that Blissidae species might have evolved with the continuous radiation of the host after the diversification of Poaceae. However, at present, the speculation about the co-evolution between Blissidae and Poaceae was very preliminary through the consistency of time in this study, and we still need more subsequent evidence from biology and molecular science to prove it. Meanwhile, Clade B contained three species, with bamboo as the host plant, indicating that its monophyly might be involved with host adaptation. Moreover, in combination with base composition analysis, we found that there were significant differences in AT skew between different clades, and its biological significance needs to be further discussed with more genomic data.

At present, there are still some problems in the reconstruction of the phylogeny of Blissidae, especially with the lack of known Blissidae molecular data. Therefore, more molecular data need to be acquired to provide a more comprehensive view of the evolution of Blissidae and Lygaeoidea. At the same time, we should pay more attention to the relevant host information of Blissidae species, and further reveal the evolutionary history between them through more molecular and biological evidence in the future.

## 4. Conclusions

In this study, 10 new mitogenomes of eight genera from Blissidae were sequenced and analyzed. Coupled with published data, phylogenetic analyses and divergence time were performed in Blissidae. A gene rearrangement with the duplication of tRNA-H was identified within *Pirkimeru japonicus*, which was the first instance of mitochondrial gene rearrangement discovered in Lygaeoidea. The divergence within Blissidae began at about 56 Ma, and the divergence at the genus level was concentrated at 51–30 Ma, slightly later than that of the host plant Poaceae. The consistency of the divergence time between Blissidae and Poaceae might hint at the coevolutionary relationship between them, but further molecular and biological evidence is still needed to prove it.

## Figures and Tables

**Figure 1 insects-13-00643-f001:**
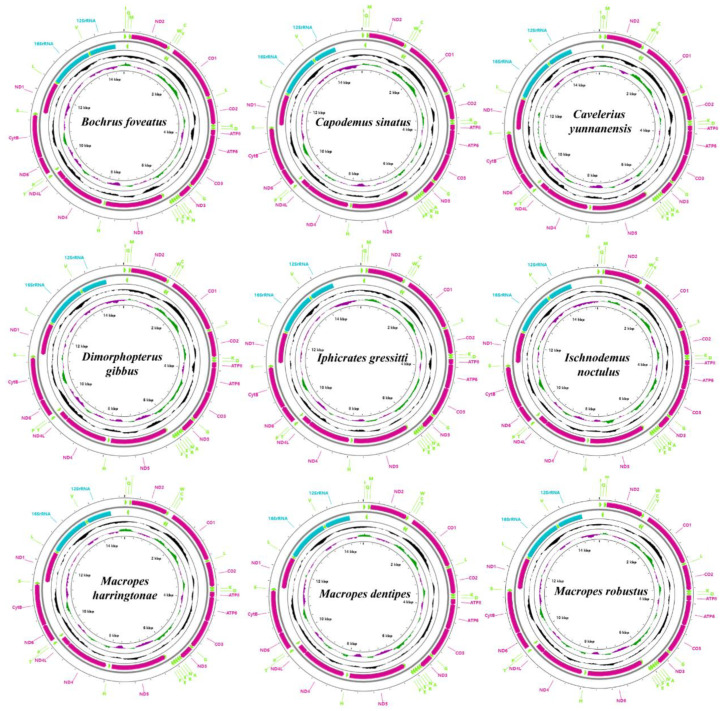
Structure of the mitogenomes’ nine species. Arrows indicate the orientation of gene transcription. Ticks in the inner circle indicate sequence length.

**Figure 2 insects-13-00643-f002:**
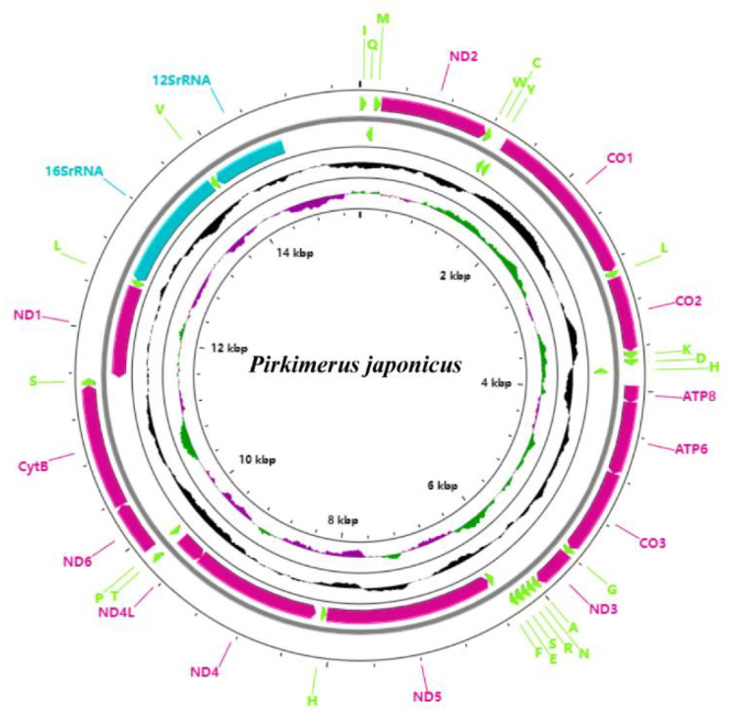
Structure of the mitogenomes of *Pirkimeru japonicus*. Arrows indicate the orientation of gene transcription. Ticks in the inner circle indicate sequence length.

**Figure 3 insects-13-00643-f003:**
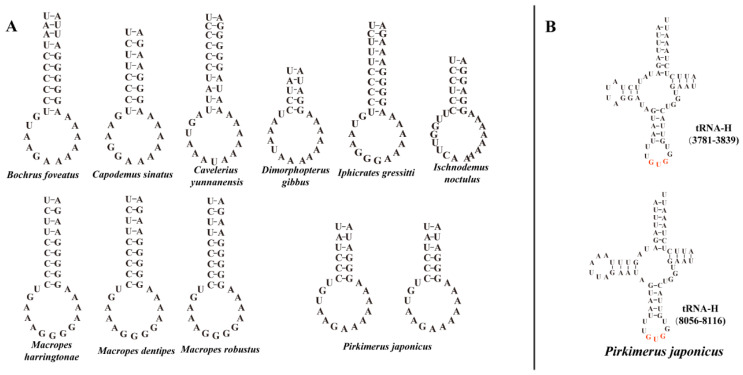
(**A**) The stem-and-loop structure located in between the tRNA-H and the ND4 from Blissidae. (**B**) Secondary structures of two tRNA-H of *Pirkimeru japonicus*. Numbers in parentheses indicate the location in the mitogenome.

**Figure 4 insects-13-00643-f004:**
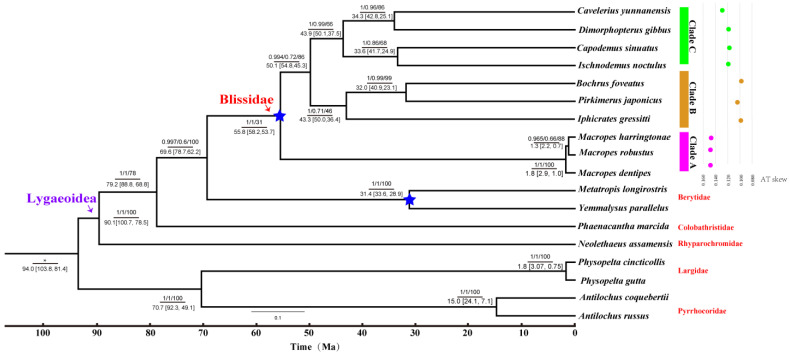
Phylogenetic tree and divergence time estimation inferred from PCG12RT. (Numbers in nodes: numbers above the line are posterior probability from MrBayes/posterior probability from Phylobayes-MPI/bootstrap from IQ-TREE, asterisk indicates that relevant parameters are not obtained; numbers below the line are divergence times; nodes marked with five pointed stars represent fossil marks.)

**Table 1 insects-13-00643-t001:** AT-content, AT-, and GC-skew of 10 Blissidae mitochondrial genomes.

Species	Sample_ID	Mitogenome Excluding Control Region				PCGs			
		Size	AT%	AT Skew	GC Skew	Size	AT%	AT Skew	GC Skew
*Bochrus foveatus*	BoYNGL1	14488	76.8	0.104	−0.207	10974	76.3	−0.119	−0.017
*Capodemus sinuatus*	CpYNRL1	14385	75.9	0.117	−0.187	10977	75.3	−0.118	−0.008
*Cavelerius yunnanensis*	CvYNLC1	14470	76.4	0.128	−0.186	10929	75.8	−0.100	−0.017
*Dimorphopterus gibbus*	DmGXBS1	14500	77.6	0.119	−0.179	10980	77	−0.104	0.017
*Iphicrates gressitti*	IpZJLA1	14459	77.9	0.099	−0.167	10944	77.7	−0.130	0.031
*Ischnodemus noctulus*	IsYNRL1	14451	75.6	0.119	−0.123	10977	74.8	−0.123	0.004
*Macropes harringtonae*	MaGZZY1	14516	74.2	0.148	−0.202	10965	73	−0.129	−0.019
*Macropes dentipes*	MaYNBN1	14514	74.2	0.148	−0.194	10965	73	−0.126	−0.026
*Macropes robustus*	MaYNGL1	14513	74.1	0.147	−0.197	10965	73	−0.129	−0.022
*Pirkimerus japonicus*	PiGXBS1	14643	77.7	0.097	−0.148	10971	77.1	−0.110	0.009
**Species**	**Sample_ID**	**rRNA**				**tRNA**			
		**Size**	**AT%**	**AT Skew**	**GC Skew**	**Size**	**AT%**	**AT Skew**	**GC Skew**
*Bochrus foveatus*	BoYNGL1	2004	79.3	0.175	−0.304	1432	76.3	0.051	−0.114
*Capodemus sinuatus*	CpYNRL1	1927	78.7	0.149	−0.290	1435	76.9	0.030	−0.091
*Cavelerius yunnanensis*	CvYNLC1	1991	77.8	0.162	−0.265	1447	78	0.072	−0.109
*Dimorphopterus gibbus*	DmGXBS1	1998	79.4	0.159	−0.275	1457	78.3	0.055	−0.115
*Iphicrates gressitti*	IpZJLA1	2006	79.7	0.162	−0.291	1426	77.1	0.045	−0.092
*Ischnodemus noctulus*	IsYNRL1	1995	78.7	0.164	−0.249	1437	77.4	0.072	−0.053
*Macropes harringtonae*	MaGZZY1	2004	78.5	0.200	−0.250	1444	76.8	0.036	−0.112
*Macropes dentipes*	MaYNBN1	2004	78.6	0.196	−0.247	1444	76.8	0.036	−0.103
*Macropes robustus*	MaYNGL1	2004	78.5	0.197	−0.250	1444	76.6	0.034	−0.106
*Pirkimerus japonicus*	PiGXBS1	2003	80	0.125	−0.240	1416	78.8	0.038	−0.070

## Data Availability

Newly sequenced mitogenomes: GenBank accession no. ON961018–ON961027.

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
