# Peer review of "Mitogenomics of Chinch Bugs from China and Implications for Its Coevolutionary Relationship with Grasses"

_insects, 2022, doi:10.3390/insects13070643_

Round 1
Reviewer 1 Report
The key problem is that the papers does not really present strong evidence for co-evolution between the insects and the host. Divergence time is used as an indicator for co-evolution - I feel that can be misleading and more detailed work is needed to show/demonstrate co-evolution between insects/host (see comments made on the pdf file). For example phylogenetic reconstruction of the hosts to be "mirrored" with the insects etc. The paper does present new data on mitogenomes for this group of insects and that is useful for the field and by focusing on the comparative aspects I think this would be a valuable contribution. "As is" The title needs to be modified and the "co evolution" aspects can be hinted/speculated at - BUT cannot be the focus of the paper. The manuscript needs careful proof reading and editing for grammar etc.
NOT sure what the policy of the journal is BUT for ->Data Availability Statement: All newly sequences genetic data will be submitted to GenBank?
I would assume Data "have to be submitted" for final acceptance!

Reviewer 2 Report
please refer to the PDF for the specific comments

Round 2
Reviewer 1 Report
The authors addressed my concerns.